# Genotype-Dependent Immunogenicity of Dengue Virus Type 2 Asian I and Asian/American Genotypes in Common Marmoset (*Callithrix jacchus)*: Discrepancy in Neutralizing and Infection-Enhancing Antibody Levels between Genotypes

**DOI:** 10.3390/microorganisms9112196

**Published:** 2021-10-21

**Authors:** Nor Azila Muhammad Azami, Meng Ling Moi, Yasushi Ami, Yuriko Suzaki, Satoshi Taniguchi, Shigeru Tajima, Masayuki Saijo, Tomohiko Takasaki, Ichiro Kurane, Chang-Kweng Lim

**Affiliations:** 1Department of Virology 1, National Institute of Infectious Diseases, Tokyo 162-8640, Japan; azila_azami@ukm.edu.my (N.A.M.A.); rei-tani@nih.go.jp (S.T.); stajima@nih.go.jp (S.T.); msaijo@nih.go.jp (M.S.); 2UKM Medical Molecular Biology Institute, Universiti Kebangsaan Malaysia, Kuala Lumpur 56000, Malaysia; 3School of International Health, The University of Tokyo, Tokyo 113-0033, Japan; sherry@m.u.tokyo.ac.jp; 4Division of Experimental Animal Research, National Institute of Infectious Diseases, Tokyo 208-0011, Japan; yami@nih.go.jp (Y.A.); ysuzaki@nih.go.jp (Y.S.); 5Kanagawa Prefectural Institute of Public Health, Kanagawa 253-0087, Japan; takasaki.jp58@pref.kanagawa.jp; 6National Institute of Infectious Diseases, Tokyo 162-8640, Japan; kurane@nih.go.jp

**Keywords:** dengue, common marmoset, neutralizing antibody, genotypes, FcγR-expressing BHK cells, infection-enhancing

## Abstract

Owing to genotype-specific neutralizing antibodies, analyzing differences in the immunogenic variation among dengue virus (DENV) genotypes is central to effective vaccine development. Herein, we characterized the viral kinetics and antibody response induced by DENV type 2 Asian I (AI) and Asian/American (AA) genotypes using marmosets (*Callithrix jacchus*) as models. Two groups of marmosets were inoculated with AI and AA genotypes, and serial plasma samples were collected. Viremia levels were determined using quantitative reverse transcription-PCR, plaque assays, and antigen enzyme-linked immunosorbent assay (ELISA). Anti-DENV immunoglobulin M and G antibodies, neutralizing antibody titer, and antibody-dependent enhancement (ADE) activity were determined using ELISA, plaque reduction neutralization test, and ADE assay, respectively. The AI genotype induced viremia for a longer duration, but the AA genotype induced higher levels of viremia. After four months, the neutralizing antibody titer induced by the AA genotype remained high, but that induced by the AI genotype waned. ADE activity toward Cosmopolitan genotypes was detected in marmosets inoculated with the AI genotype. These findings indicate discrepancies between heterologous genotypes that influence neutralizing antibodies and viremia in marmosets, a critical issue in vaccine development.

## 1. Introduction

Dengue virus (DENV) belongs to the genus *Flavivirus* in the family *Flaviviridae*. DENV is a small (~11 kb) enveloped virus that contains a single-stranded, positive-sense RNA genome [1,2,3]. The RNA genome of DENV is formed by a single open reading frame that encodes three structural proteins (capsid [C], pre-membrane [prM], and envelope [E]) and seven non-structural proteins (NS, NS1, NS2a, NS2b, NS3, NS4a, NS4b, and NS5) [3,4,5]. There are four antigenically distinct serotypes of DENV, referred to as DENV1–4, with the identity among serotypes being less than 80% at their E protein amino acid level [5,6,7]. Nonetheless, infection with any serotype of DENV causes similar clinical symptoms ranging from mild febrile illness, dengue without warning signs, dengue with warning signs, severe dengue, and occasionally dengue-related death. Symptoms in patients diagnosed as dengue with warning signs include abdominal pain, persistent vomiting, fluid accumulation, mucosal bleeding, lethargy, liver enlargement, increased hematocrit with a decrease in platelets [8]. In 2009, the World Health Organization classified dengue cases as severe dengue case symptoms, including severe plasma leakage, with or without severe hemorrhagic, and severe organ impairment including myocarditis, hepatitis, and encephalitis [9,10,11,12].

DENV is transmitted by *Aedes* mosquitoes and the virus infects up to 390 million people per year, of which 96 million infections demonstrate a spectrum of clinical severity [13]. DENV type 2 (DENV2) is the most frequent cause of dengue epidemics worldwide [14,15]. Phylogenetic analysis of the E protein revealed six genotypes of DENV2: sylvatic, Cosmopolitan (CM), Asian I (AI), Asian II, Asian/American (AA), and American [15,16]. Although genotype differences in genome sequences are approximately 3% to 6%, they differ in virulence, incidence, and vector competence [17,18,19,20,21]. In-depth analyses of the immunogenicity of each genotype are central to the development of an effective dengue vaccine. Sequence variations within each genotype strain affects the ability of a vaccine to induce antibodies to neutralize all strains with the homologous serotype because neutralizing antibodies are highly genotype-specific [22]. Due to these serotype-specific neutralizing antibody properties, cross-reactive neutralization ability against heterologous DENV serotypes wanes after a few months [2]. Nonetheless, these subneutralizing antibodies facilitate virus entry into the cells via Fc gamma receptor (FcγR) and subsequently enhance the viral infection and lead to an increase in viral load [23,24,25]. Dengvaxia is a tetravalent live-attenuated chimeric DENV vaccine and is implemented in a three-dose series at six-month intervals (at months 0, 6, and 12). However, its efficacy against all DENV serotypes ranges from 44.6% to 65.6% [26,27]. Notably, the efficacy of this vaccine against DENV2 is the lowest compared to that against other serotypes. It has been hypothesized that low efficacy is probably associated with strain variation among DENV genotypes [22,27].

We previously reported that heterogeneity and homogeneity of the infecting genotypes influence the levels and cross-reactivity of neutralizing antibodies induced after primary, secondary, and tertiary infections [28]. Defining the variation in viremia induction, antibody response pattern, and antibody-dependent enhancement (ADE) activity in different genotypes of DENV is essential for understanding dengue pathogenesis and developing effective vaccines. Common marmosets (*Callithrix jacchus*) are useful animal models of DENV infection [29,30]. Thus, this study aimed to characterize in detail the differences between DENV2 genotypes in terms of viremia levels, neutralizing antibody levels, and ADE activity using common marmosets that were inoculated in a two-dose series at four-month intervals (at months zero and four) with a homological genotype of DENV2. Elucidating the immunological ability, induced by two homological genotype infections in a four-month interval, toward the different genotypes of DENV2 could clarify the important immunological characterization of each genotype for vaccine development.

## 2. Materials and Methods

### 2.1. Viruses

DENV type 1 (DENV1) Tokyo-Yoyogi (GenBank accession no. LC006123), DENV2 DHF0663 (AB189122), DENV2 00-43 (AB111452), DENV2 08-77 (AB545874), DENV type 3 (DENV3) NRT 09-34, DENV type 4 (DENV4) TVP360, Zika virus (ZIKV) PRVABC59 (KX377337), and Japanese encephalitis virus (JEV) Mie/41/2002 (AB241119) strains were used. The DENV2 00-43 strain was isolated from imported dengue fever cases from Indonesia and belonged to the AI genotype. The DENV2 08-77 strain was isolated from an imported dengue fever case from the Maldives and belonged to the AA genotype. DENV2 DHF006 was isolated from a dengue hemorrhagic fever case that belonged to the Cosmopolitan genotype. All virus strains were first isolated using C6/36 cells and passaged in baby hamster kidney (BHK) cells. Virus stocks were used within four culture passages in BHK cells. Culture supernatants collected from infected BHK cells were centrifuged at 3000 rpm for 5 min to remove cell debris. The virus stock was stored at −80 °C before use. The virus RNA was purified and sequenced as described previously [31]. A phylogenetic tree was constructed using the nucleotide sequence of the complete E protein region and obtained using the neighbor-joining method in MEGA software [32].

### 2.2. DENV Infection in Marmoset Models

Common marmosets (*Callithrix jacchus*) were obtained from CLEA Japan, Inc. (Tokyo, Japan) and maintained in a specific pathogen-free area at the National Institute of Infectious Diseases (NIID, Tokyo, Japan). The NIID Animal Study Review Board approved this study (approval nos. 613006 and 516010). Each marmoset was inoculated with 1 × 10^6^ plaque-forming units (PFU/dose of DENV). Marmosets M1, M2, and M3 were inoculated with the DENV2 00-43 strain (AB111452), and Marmosets M4, M5, and M6 were inoculated with the DENV2 08-77 strain (AB545874). The interval between DENV inoculations was 140 days. In total, 1 mL of whole blood was collected in EDTA tubes from each marmoset on day 0 (before virus inoculation), 2, 4, 7, 10, 14, 114, 128, and 140 post-inoculation (p.i.) following the first virus inoculation, and on days 2, 4, 7, 10, 14, and 148 p.i. following the second virus inoculation. The blood samples were centrifuged at 2000 rpm for 10 min at 4 °C to collect the plasma. The plasma samples were stored at −80 °C before use.

### 2.3. Determination of Anti-DENV Immunoglobulin M (IgM) and IgG Antibody

Levels of anti-DENV IgM antibody against all DENV serotypes were determined using a Focus Dengue Fever IgM capture ELISA kit (Focus Diagnostic, Cypress, CA, USA). Anti-DENV IgG antibodies against all serotypes of DENV were determined using a Panbio Dengue IgG Indirect ELISA kit (Abbott Laboratories, Abbott Park, IL, USA). Plasma samples from three DENV-naïve marmosets were used as negative controls. A positive: negative ratio was calculated using the following formula: (index value of sample/index value of negative controls). A positive: negative ratio (P: N ratio) ≥2 was considered positive. Index values were calculated according to the manufacturer’s instructions.

### 2.4. Determination of Neutralizing Antibody Titers Using a 50% Plaque Reduction Neutralization Test Assay (PRNT_50_)

BHK cells were seeded in 12-well plates in Minimum Essential Medium (MEM) (Merck Millipore, Burlington, MA, USA) supplemented with 10% heat-inactivated fetal bovine serum (Hi-FBS) (Thermo Fisher Scientific, Waltham, MA, USA) and then incubated at 37 °C overnight until the cell monolayer reached 70%–80% confluency. Plasma samples were heat-inactivated at 56 °C for 30 min before use. The heat-inactivated plasma samples were serially diluted two-fold, starting from 1:2.5 to 1:20,480 in MEM supplemented with 10% Hi-FBS. The virus-antibody complexes were prepared by mixing 25 μL of DENV at titers of 5000 PFU/mL with 25 μL serially diluted plasma samples to make a final dilution series from 1:5 to 1:40,960. The control samples were prepared by mixing 25 μL of DENV at 5000 PFU/mL with 25 μL of MEM supplemented with 10% Hi-FBS. Next, the virus-antibody complexes were incubated at 37 °C for 60 min. A total of 50 μL of the mixture was then inoculated onto BHK cell monolayers in 12-well plates. After incubation for 60 min, 1 mL of maintenance medium containing MEM and 1% methylcellulose supplemented with 2% Hi-FBS was added. The plates were incubated at 37 °C in 5% CO_2_ until visible plaques were observed (5–7 days of incubation). Cells were fixed with 10% formaldehyde, stained with methylene blue, and washed with tap water, after which the number of plaques was counted. All tests were conducted in duplicate. The neutralizing antibody titer was expressed as the maximum dilution of the plasma sample, which yielded a ≥50% plaque reduction in the virus inoculum compared to that in the control virus samples. Results are shown as PRNT_50_ values, expressed as the reciprocal of the highest plasma dilution (end-point titer), resulting in ≤50% of the input plaque count. Differences in the neutralizing antibody titers were determined by comparing the geometric mean (GMT) of genotype-specific neutralizing antibody titers of DENV2 AI, CM, and AA genotypes at each time point.

### 2.5. Determination of Infection-Enhancing Activity Using the ADE Assay with BHK-21 and FcγR-Expressing BHK Cells

The presence of enhancing activity was determined in plasma samples collected at 140 days after the first virus inoculation using an ADE assay, as described previously [33]. Plasma samples were heat-inactivated at 56 °C for 30 min before use. The heat-inactivated plasma samples were serially diluted 10-fold, starting from 1:10 to 1:10^6^ in MEM supplemented with 10% Hi-FBS. The fold-enhancement was calculated using the following formula: virus titers in plasma sample/virus titers in the absence of plasma samples (negative control). To assess the contribution of the virus-immune complex to virus titers (infection-enhancement), the ratios of fold-enhancement were calculated using the following formula: average fold-enhancement assessed with FcγR-expressing BHK cells (V_N_)/average virus fold-enhancement assessed with BHK cells (V_O_) [34].

### 2.6. Viral RNA Extraction

DENV RNA (vRNA) was extracted from 200 µL of plasma using a High Pure Viral RNA kit (Roche Diagnostics, Basel, Switzerland). The final volume was 50 µL of extracted RNA per sample.

### 2.7. Quantification of Viral RNA Copy Number

Viral RNA copy number was determined using quantitative TaqMan real-time reverse transcriptase-polymerase chain reaction (qRT-PCR) using primers and reagents described previously [35]. Viral genome levels were expressed as log10 genome copies per milliliter.

### 2.8. Determination of DENV Titer Using Plaque Assay

Baby hamster kidney cell line-21 (BHK-21) was seeded in 12-well plates in MEM (Merck Millipore) supplemented with 10% Hi-FBS (Thermo Fisher Scientific) and then incubated at 37 °C overnight until the cells reached approximately 70% confluency. The plasma samples were serially diluted 10-fold from 1:10 to 1:10^6^. A total of 50 µL of diluted plasma was inoculated onto the cell monolayer. After incubation for 60 min, 1 mL of maintenance medium containing MEM and 1% methylcellulose supplemented with 2% Hi-FBS was added. The plates were incubated at 37 °C in 5% CO_2_ until visible plaques were observed (5–7 days of incubation). Plaques were fixed with 10% formaldehyde, stained with methylene blue, washed with water, and counted. All tests were conducted in duplicate. Viral titers were expressed as plaque-forming units per milliliter (PFU/mL) using the following formula: (average number of plaques observed × sample dilution)/inoculum volume, µL).

### 2.9. Determination of Dengue NS1 Antigen

Dengue NS1 antigen levels were determined using a Platelia™ Dengue NS1 Ag ELISA kit (Bio-Rad Laboratories, Inc., Hercules, CA, USA). Plasma samples from three DENV-naïve marmosets were used as negative controls. A positive: negative ratio was calculated using the following formula: (index value of sample/index value of negative controls). A positive: negative ratio (P: N ratio) ≥2 was considered positive, and all tests were performed in duplicates.

### 2.10. Data Analysis

Data were analyzed using the statistical analysis tool pack in Microsoft Excel 2016 (Microsoft Corporation, Redmond, WA, USA) and GraphPad Prism 8 (GraphPad Software Inc., San Diego, CA, USA). Student’s t-test was used in the analysis, and *p* values < 0.05 were considered statistically significant.

### 2.11. Ethical Approval

Animal studies were conducted in accordance with the guidelines of the Institutional Animal Care and Use Committee of the NIID, Tokyo, Japan. The study was approved by the Institutional Animal Care and Use Committee of NIID (approval nos. 613006 and 516010) on 29 January and 7 March 2013, respectively. All animal and infection experiments were performed in accordance with NIID Institutional.

## 3. Results

### 3.1. Viremia Levels

Six marmosets were used in the study. Marmosets numbered 1 to 3 (M1, M2, and M3) were inoculated with the DENV2 Asian I (AI) genotype. Marmosets 4 to 6 (M4, M5, and M6) were inoculated with the DENV2 Asian/American (AA) genotype. NS1 levels, infectious virus titers, and viral RNA copy numbers were determined on days 0, 2, 4, 7, 10, and 14 p.i. (Figure 1). The mean duration of the NS1 antigen detected in the marmosets inoculated with the DENV2 AI genotype was longer (average detection period = 7.0 ± 1.7 days) than that in marmosets inoculated with the DENV2 AA genotype (average detection period = 6.0 ± 0.0 days) (Figure 1a). After the second virus inoculation, the levels of NS1 antigen were below the detection level due to the presence of antibodies specific to NS1 antigen. Nonetheless, there were no significant differences in the levels of NS1 antigen between marmosets inoculated with DENV2 AI and AA genotypes in the first and second virus inoculation.

In the marmosets inoculated with the DENV2 AI genotype, the average detection period of the infectious virus after the first virus inoculation was 3.0 ± 1.7 days, with average peak levels of (3.4 ± 2.1) × 10^3^ PFU/mL (Figure 1b). In the marmosets inoculated with the DENV2 AA genotype, the average detection period of the infectious virus was 2.0 ± 0.0 days, with average peak levels of (8.2 ± 5.0) × 10^3^ PFU/mL. Nonetheless, we found no significant differences in infectious virus titers between marmosets inoculated with the DENV2 AI and AA genotypes.

Viral RNA was detected on day 2 p.i. in all marmosets inoculated with the DENV2 AI and AA genotypes after the first virus inoculation (Figure 1c). In the marmosets inoculated with the DENV2 AI genotype, the average detection period of viral RNA was 3.6 ± 2.9 days, with average peak levels of 3.82 ± 0.9 log10 genome copies/mL. In the marmosets inoculated with the DENV2 AA genotype, the average detection period of viral RNA was 7.3 ± 1.2 days, with average peak levels of 3.97 ± 0.2 log10 genome copies/mL. Interestingly, the levels of viral RNA in marmosets inoculated with the AA genotype were significantly higher than those in marmosets inoculated with the A1 genotype at day 7 p.i. (*p* = 0.009) and day 10 p.i. (*p* = 0.001).

### 3.2. Anti-DENV IgM and IgG Antibody Levels

All marmosets were inoculated with the homologous genotype of DENV2 during the secondary infection to determine the antibody pattern following infection. The levels of anti-DENV IgM and IgG antibodies were determined on days 0 (before the first virus inoculation), 2, 4, 7, 10, 14, 114, 128, and 140 p.i. following the first virus inoculation, and on days 2, 4, 7, 10, and 14 following the second virus inoculation (Figure 2). Samples could not be obtained from marmoset M3 owing to non-study-related death at day 128 following virus inoculation; marmoset M3 demonstrated no symptoms related to infectious diseases, and the autopsy remained inconclusive regarding the reason for death.

Following primary virus inoculation, anti-DENV IgM antibody was first detected after day 4 p.i. and earlier than anti-DENV IgG antibody in the marmosets inoculated with DENV AI genotype (Figure 2a). Anti-DENV IgG antibody levels were first detected after day 10 p.i. After three months, anti-DENV IgM antibody levels waned below detection, but anti-DENV IgG antibody levels continued to increase. Following the second virus inoculation with the DENV2 AI genotype, anti-DENV IgM levels remained below the detection level, but high levels of anti-DENV IgG antibody were detected in the marmosets.

In marmosets inoculated with the DENV2 AA genotype, anti-DENV IgM and IgG antibodies were first detected at day 7 and 10 p.i., respectively (Figure 2b). Before secondary virus inoculation, high anti-DENV IgG antibody levels were still detected in all marmosets, but anti-DENV IgM antibodies were below the detection level. Following the second virus inoculation, anti-DENV IgM antibody levels slightly increased, but the level of anti-DENV IgG continued to increase. Both the DENV2 AI and AA genotypes induced similar antibody responses in marmosets. High levels of anti-DENV IgG antibodies were induced in marmosets inoculated with DENV2 AI and AA genotypes following the second virus inoculation.

### 3.3. Levels of Neutralizing Antibody to DENV2 Asian I, Cosmopolitan, and AA Genotypes

Levels of genotype-specific and genotype cross-reactive neutralizing antibodies to three genotypes of DENV2 (AI, CM, and AA) were determined using a PRNT_50_ on day 0 (before virus inoculation), 4, 7, 14, 114, 128, and 140 p.i. following the first virus inoculation and on days 2, 4, 7, 14, and 148 p.i. following the second virus inoculation (Figure 3).

In the marmosets inoculated with the DENV2 AI genotype, neutralizing antibodies to the DENV2 AI, CM, and AA genotypes were detected on day 14 p.i. (GMT_CM_ = 40, GMT_AI_ = 80, GMT_AA_ = 160) (Figure 3a). Four months after the first virus inoculation, the levels of neutralizing antibodies to the DENV2 AI, CM, and AA genotypes waned (GMT_CM_ < 20, GMT_AI_ < 20, GMT_AA_ = 25). Following the second virus inoculation with the DENV2 AI genotype, neutralizing antibody levels in the marmosets increased rapidly on day 7 p.i. Levels of neutralizing antibodies against the DENV2 AA genotype (GMT_AA_ = 320) were higher than those of neutralizing antibodies to the DENV2 CM (GMT_CM_ = 57) and DENV2 AI genotypes (GMT_AI_ = 113). On day 14 p.i., levels of neutralizing antibodies to the DENV2 AI, CM, and AA genotypes increased approximately 4- to 8-fold compared to those during primary virus inoculation, and levels of neutralizing antibodies to the DENV2 AA genotype were higher than those of neutralizing antibodies to the DENV2 CM and AI genotypes (GMT_AI_ = 453, GMT_CM_ = 320, GMT_AA_ = 1280). Four months after the second virus inoculation, neutralizing antibodies to the DENV2 AI, CM, and AA genotypes waned (GMT_AI_ = 28, GMT_CM_ = 20, GMT_AA_ = 160) but remained detectable.

In the marmosets inoculated with the DENV2 AA genotype, neutralizing antibodies to the DENV2 AI, CM, and AA genotypes were detected on day 14 p.i. (GMT_AI_ = 50, GMT_CM_ = 14, GMT_AA_ = 63) (Figure 3b). Four months after the first virus inoculation, levels of neutralizing antibodies to the DENV2 AI, CM, and AA genotypes continued to increase and remained high (GMT_AI_ = 80, GMT_CM_ = 160, GMT_AA_ = 403). Following the second virus inoculation, levels of neutralizing antibodies to the DENV2 AI, CM, and AA genotypes increased up to 10-fold compared to those on day 14 p.i. following the first virus inoculation (GMT_AI_ = 640, GMT_CM_ = 508, GMT_AA_ = 4063). After four months, although the levels of neutralizing antibodies decreased, the antibodies remained detectable (GMT_AI_ = 28, GMT_CM_ = 80, GMT_AA_ = 320).

### 3.4. Cross-Reactive Neutralizing Antibodies and Infection-Enhancing Activity to Heterologous Genotype and Serotype

Levels of genotype and serotype cross-reactive neutralizing antibodies to DENV1, DENV2 AI, DENV2 CM, DENV2 AA, DENV3, and DENV4 plasma at 4 months after the first virus inoculation were determined using a PRNT_50_ assay (Table 1). In the marmosets inoculated with the DENV2 AI genotype, serotype cross-reactive neutralizing antibodies against DENV1, DENV3, and DENV4 were below the detection level. Conversely, low-genotype cross-reactive neutralizing antibodies against the DENV2 CM and AA genotypes were detected (GMT_CM_ < 5, GMT_AA_ = 28). In the marmosets inoculated with the DENV2 AA genotype, low serotype cross-reactive neutralizing antibodies against DENV1, DENV3, and DENV4 were detected (GMT_DENV1_ = 4, GMT_DENV3_ = 8, GMT_DENV4_ = 6), and high levels of genotype cross-reactive neutralizing antibodies to the DENV2 AI and CM genotypes were detected (GMT_AI_ = 160, GMT_CM_ = 253).

The presence of infection-enhancement activity to heterologous serotypes and genotypes (DENV1, DENV2 CM, and DENV3) was determined in the marmosets inoculated with the DENV2 AI and AA genotypes using an ADE assay (Figure 4).

The virus titers of DENV1 were higher with the mean infection-enhancement of 2.3-fold in the presence of 1:10 diluted plasma from the marmosets inoculated with DENV2 AI genotype than in the absence of plasma in FcγR-expressing BHK cells (Figure 4a). In marmosets inoculated with the DENV2 AA genotype, the infection-enhancement was 2-fold higher in FcγR-expressing BHK cells (Figure 4b). In marmosets inoculated with DENV2 AI genotype, the virus titers of the DENV2 CM genotype were higher with a mean infection-enhancement of 2.7-fold in the presence of 1:100 diluted plasma than in the absence of plasma when inoculated in the FcγR-expressing BHK cells (Figure 4c). In marmosets inoculated with DENV2 AA genotype, the titer of DENV2 was higher with a mean infection-enhancement of 2-fold in the presence of 1:1000 diluted plasma than in the absence of plasma when inoculated in the FcγR-expressing BHK cells (Figure 4d). Interestingly, in the BHK cells, the virus titers of the DENV2 CM genotype were lower in the presence of plasma than in the absence of plasma from the marmosets inoculated with DENV2 AI and AA genotypes. In marmosets inoculated with DENV2 AI genotype, the virus titers of DENV3 were higher, with a mean infection-enhancement 2- and 2.7-fold in the presence of 1:10 diluted plasma than in the absence of plasma in BHK-and FcγR-expressing BHK cells, respectively (Figure 4e). In marmosets inoculated with DENV2 AA genotype, the virus titers of DENV3 were higher with a mean infection-enhancement 2-fold in the presence of 1:100 diluted plasma than in the absence of plasma in BHK cells (Figure 4f).

The ratio of fold infection-enhancement in FcγR-expressing BHK cells, V_N_/fold infection-enhancement in BHK cells, and V_O_ was calculated to assess the infection-enhancement activity due to the presence of virus-antibody complexes (Figure 5).

In the marmosets inoculated with the DENV2 AI and AA genotypes, the ratio of infection-enhancement to DENV1 was less than 4-fold, suggesting that ADE activity to DENV1 was absent (Figure 5a). Interestingly, in the marmosets inoculated with the DENV2 AI genotype, the mean fold ratio of infection-enhancement to DENV2 CM genotype at plasma dilutions of 1:10 and 1:100 was 27- and 4-fold, respectively, suggesting the presence of an infection-enhancement antibody to the DENV2 CM genotype (Figure 5b). In the marmosets inoculated with the DENV2 AA genotype, the ratio of infection-enhancement to DENV2 CM was less than 4-fold, suggesting that ADE activity to DENV2 was absent (Figure 5b). In the marmosets inoculated with the DENV2 AI and AA genotypes, the ratio of infection-enhancement to DENV3 was less than 4-fold, suggesting that ADE activity to DENV3 was absent (Figure 5c).

### 3.5. Cross-Reactive Neutralizing Antibodies to ZIKV and JEV

Levels of cross-reactivity antibodies against ZIKV and JEV 4 months post-inoculation were determined using a PRNT_50_ assay (Table 2). Cross-reactive neutralizing antibodies against ZIKV and JEV were absent in marmosets inoculated with the DENV2 AI and AA genotypes. However, a low neutralizing antibody against JEV was detected in one marmoset inoculated with the DENV2 AA genotype.

## 4. Discussion

In a previous study, we reported that varying levels of neutralizing antibodies were induced against different DENV2 genotypes after primary, secondary, and tertiary infections and that the neutralizing antibody titers to some heterologous genotypes were higher than those of homologous genotypes within a serotype [29]. Thus, this study analyzed the differences in the immunogenic variation among genotype-specific neutralizing antibodies against DENV by characterizing the viral kinetics and antibody response induced by DENV2 AI and AA genotypes using marmosets as the animal model.

Here, we found that the DENV2 AI and AA genotypes induced different levels of viremia and neutralizing antibody titers in marmosets. Inoculation with the DENV2 AI and AA genotypes induced different levels of genotype-specific neutralizing antibodies against the DENV2 AI, CM, and AA genotypes. On day 14 p.i., following the first virus inoculation, higher levels of genotype-specific neutralizing antibodies against the DENV2 AI, CM, and AA genotypes were detected in marmosets inoculated with the DENV2 AI genotype than in those inoculated with the DENV2 AA genotype. However, at four months p.i., neutralizing antibodies in the marmosets inoculated with the DENV2 AA genotype remained high and detectable compared to those in the marmosets inoculated with the DENV2 AI genotype, which decreased. Neutralizing antibodies in marmosets inoculated with the DENV2 AA genotype had a higher affinity for the homologous genotype. However, neutralizing antibodies in marmosets inoculated with the DENV2 AI genotype had a higher affinity for heterologous genotypes (CM and AA genotypes).

Almost 5 months (~140 days) after the first virus inoculation, ADE activity to DENV1 and DENV3 and serotype cross-reactive neutralizing antibodies against DENV1, DENV3, and DENV4 were not detected in the marmosets inoculated with the DENV2 AI genotype. However, in marmosets inoculated with the DENV2 AA genotype, low serotype cross-reactive neutralizing antibodies against DENV1, DENV3, and DENV4 were detected to enhance DENV1 and DENV3. Although the cross-reactive antibodies retained low levels of virus-neutralizing activity (≤5), the ADE activity levels were below the detection limits of the assay.

The presence of ADE activity in heterologous serotypes (DENV1 and DENV3) was not observed in any marmoset. Interestingly, ADE activity to the heterologous genotype of DENV2 (Cosmopolitan genotype) was detected in the plasma of marmosets inoculated with DENV2 AI (at 1:10 and 1:100 dilution). These results suggest that the antibodies induced by the DENV2 AI did not neutralize the heterologous genotype in FcγR-expressing BHK cells and that a low concentration of cross-reactive neutralizing antibodies is correlated with the presence of infection-enhancing activity. In addition, the results indicated that a certain threshold of cross-reactive neutralizing antibodies to heterologous genotypes and serotypes is critical for protection. Additionally, determining the spectrum of neutralizing antibodies induced by each selected DENV strain associated with protection and ADE activity is crucial for dengue vaccine development.

This study revealed significant differences in viral growth patterns between the DENV2 AI and AA genotypes in the marmoset model. Infection with the DENV2 AI genotype induced a longer detectable duration of viremia than infection with the DENV2 AA genotype in the marmoset model. Interestingly, infection with the DENV2 AA genotype induced higher levels of viremia than the DENV2 AI genotype. A study on pediatric dengue cases in Vietnam found that infection with the DENV2 AI genotype induced higher levels of viremia in children than those infected with the DENV2 AA genotype [20]. However, in this marmoset model, we found that inoculation with the DENV2 AA genotype induced higher levels of viremia than inoculation with the DENV2 AI genotype. A plausible explanation for these differences is that the DENV2 strains used in this study differed from the circulating strains in Vietnam, indicating possible discrepancies in phenotypes and pathogenicity. In pediatric dengue cases in Vietnam, clinical background, infection history, and host factors, including immune responses and genetic background, are associated with overall clinical outcomes, thus contributing to differential outcomes between human and marmoset models. Overall, our results confirm differential infectivity between genotypes, and this may, in turn, lead to distinct immune responses upon infection with different genotypes and viral strains.

This study highlights the discrepancies between genotype strains in inducing DENV-neutralizing antibodies and viremia in marmosets. The results indicate that antigenic differences between genotypes may influence the levels of serotype-specific and cross-reactive neutralizing antibodies during primary and secondary infections. Interestingly, low levels of cross-reactive neutralizing did not thoroughly neutralize the heterologous genotype in FcγR-expressing BHK cells, and ADE activity was confirmed in these samples. The results indicate the importance of optimizing the selection of DENV strains that possess the capability to neutralize homologous and heterologous genotypes. Additionally, the results indicate that neutralizing activity titers, as determined by the FcγR-expressing BHK cells, detect both infection-enhancing and neutralizing activity as neutralizing titers may better reflect the biological properties of DENV antibodies in vivo.

## 5. Conclusions

The present study demonstrated that different genotypes of DENV2 induce varying viremia levels and neutralizing antibody titers in marmosets. Antigenic differences may cause differences in the induction of viremia and antibody responses between heterologous genotypes. Our study indicated that a low concentration of heterotypic neutralizing antibodies induced by heterologic genotype strains might enhance the infection rather than protect against the infection, which may be a critical concern in vaccine development. Infection-enhancing activities of the heterologous genotypes detected in neutralizing antibodies hampered the overall neutralizing activity against DENV, as determined by the Fcγ R-expressing BHK cells. To overcome these immunological properties of genotype-specific antibodies used in vaccine development, it is necessary to identify alternative DENV strains that can induce a comprehensive set of neutralizing antibodies to effectively neutralize a wide range of DENV strains within a serotype. Furthermore, the findings of this study indicated that the marmoset model of DENV infection in combination with an ADE assay could be used to assess the efficacy of vaccine candidates.

## Figures and Tables

**Figure 1 microorganisms-09-02196-f001:**
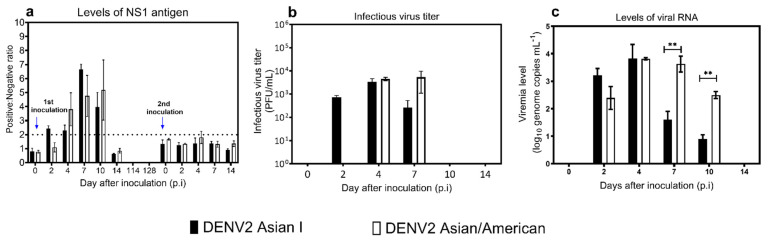
Levels of viremia in marmosets inoculated with dengue virus type 2 (DENV2) Asian I and Asian/American genotypes following virus inoculation. (**a**) Levels of NS1 antigen following the first and second virus inoculations. (**b**) Infectious virus titers (PFU/mL) following the first virus inoculation. (**c**) RNA copy number (log10 copy number/mL) following the first virus inoculation. The dashed line indicates the baseline of the positive value for the positive/negative ratio. The black bars indicate marmosets inoculated with the DENV2 Asian I genotype. The white bars indicate marmosets inoculated with the DENV2 Asian/American genotype. The error bars represent the mean ± standard error of the mean. Asterisks (**) indicate significant differences in levels of viral RNA with *p* values < 0.05.

**Figure 2 microorganisms-09-02196-f002:**
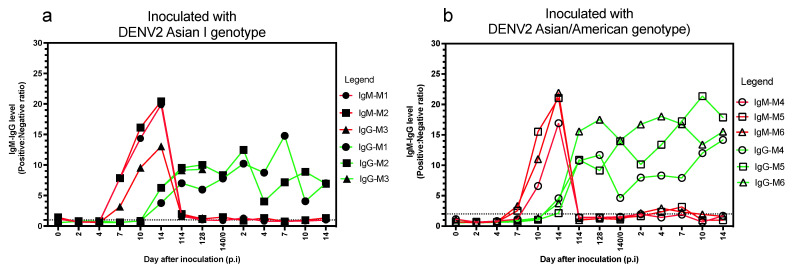
Anti-dengue virus (DENV) immunoglobulin M (IgM) and anti-immunoglobulin G (IgG) antibody levels in marmosets inoculated following virus inoculation. Anti-DENV IgM and anti-DENV IgG antibody levels in (**a**) marmosets inoculated with DENV type 2 Asian I genotype (AI) and (**b**) marmosets inoculated with DENV2 Asian/American genotype (AA). Dashed lines indicate the baseline of the positive value for the positive: negative ratio. Day 0 was defined as the day of virus inoculation. Owing to non-study-related death at day 128 following virus inoculation, samples could not be obtained from marmoset M3 beyond that day.

**Figure 3 microorganisms-09-02196-f003:**
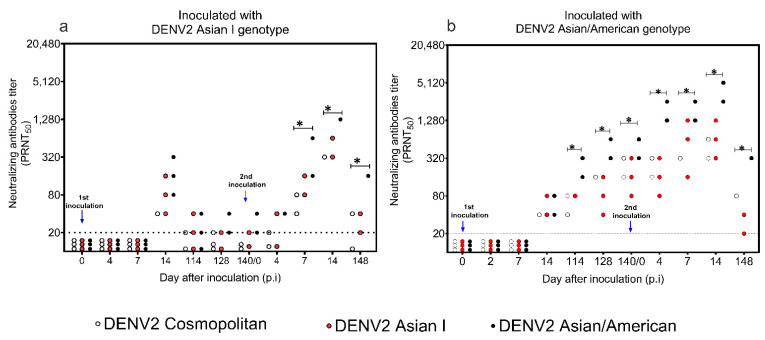
Levels of neutralizing antibodies to dengue virus type 2 (DENV2) Cosmopolitan (CM), Asian I (AI), and Asian/American (AA) genotypes following virus inoculation. Levels of neutralizing antibodies in (**a**) marmosets inoculated with the DENV2 AI genotype, and (**b**) marmosets inoculated with the DENV2 AA genotype. Asterisks (*) indicate significant differences (≥4-fold) in geometric mean titers of neutralizing antibodies among genotypes.

**Figure 4 microorganisms-09-02196-f004:**
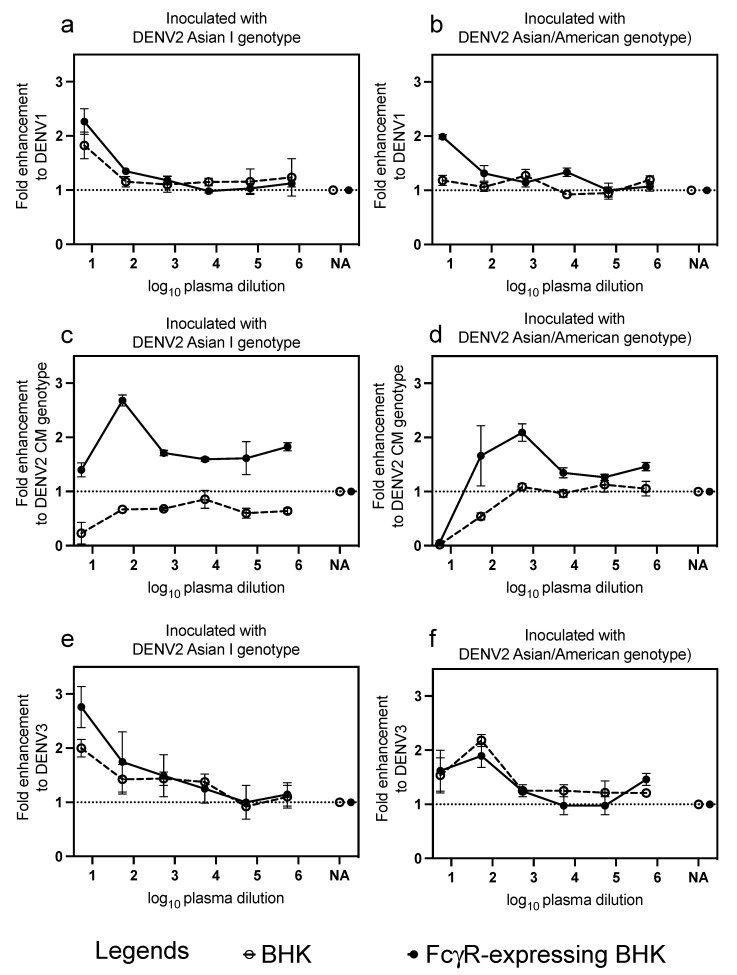
Fold infection-enhancement as determined using baby hamster kidney (BHK)-21 and FcγR-expressing BHK cells in plasma from marmosets inoculated with DENV2 AI and AA genotypes after day 140 post-inoculation. Fold infection-enhancement to DENV1 in (**a**) marmosets inoculated with DENV2 AI genotype and (**b**) marmosets inoculated with DENV2 AA genotype. Fold infection-enhancement to the DENV2 Cosmopolitan genotype in (**c**) marmosets inoculated with DENV2 AI genotype and (**d**) marmosets inoculated with DENV2 AA genotype. Fold infection-enhancement to DENV3 in (**e**) marmosets inoculated with DENV2 AI genotype and (**f**) marmosets inoculated with DENV2 AA genotype. NA indicates negative control (without the presence of plasma marmosets). The white circle indicates plasma inoculated in BHK-21 cells, and the black circle indicates plasma inoculated in FcγR-expressing BHK cells. The error bars represent the mean ± standard deviation of the mean. Due to the limited volume of samples, we were not able to test the infection-enhancing activities to DENV4.

**Figure 5 microorganisms-09-02196-f005:**
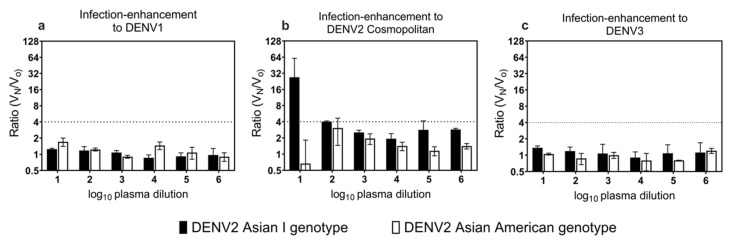
The ratio of fold infection-enhancement to heterologous serotypes and genotype using FcγR-expressing baby hamster kidney (BHK) cells. Mean ratio of infection-enhancement to (**a**) DENV1; (**b**) DENV2 Cosmopolitan genotype; and (**c**) DENV3. Dashed lines indicate an infection-enhancement fold ratio of more than 4-fold. Black bars indicate marmosets inoculated with the DENV2 Asian I genotype. White bars indicate marmosets inoculated with the DENV2 Asian/American genotype. The error bars represent the mean ± standard error of the mean. Due to the limited volume of samples, we could not determine the ratio of fold infection-enhancement to DENV4.

**Table 1 microorganisms-09-02196-t001:** Levels of neutralizing antibodies to dengue virus type 1 (DENV1), dengue virus type 2 (DENV2) Asian I (AI), DENV2 Cosmopolitan (CM), DENV2 Asian/American (AA), dengue virus type 3 (DENV3), and dengue virus type 4 (DENV4) in plasma samples of marmosets inoculated with either the DENV2 Asian I or Asian/American genotype at day 140 post-infection.

Marmoset ID	Levels of Neutralizing Antibody (PRNT_50_)					
	DENV1	DENV2 AI	DENV2 CM	DENV2 AA	DENV3	DENV4
Inoculated with DENV2 Asian I						
M1	<5	5	<5	20	5	<5
M2	<5	20	5	40	<5	<5
Inoculated with DENV2 Asian/American						
M4	<5	80	160	1280	5	10
M5	5	160	320	640	10	<5
M6	5	320	320	640	10	10

**Table 2 microorganisms-09-02196-t002:** Levels of neutralizing antibodies to Zika virus (ZIKV) and Japanese Encephalitis virus (JEV) in plasma samples from marmosets four months after inoculation with the dengue virus type 2 (DENV2) Asian I and Asian/American genotypes.

Marmoset ID	Levels of Neutralizing Antibodies (PRNT_50_)	
	ZIKV	JEV
Inoculated with DENV2 Asian I		
M1	<5	<5
M2	<5	<5
Inoculated with DENV2 Asian/American		
M4	<5	<5
M5	<5	<5
M6	<5	5

## Data Availability

The data presented in this study are available on request from the corresponding author. The data are not publicly available due to ethical guidelines.

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
