# Peer review of "Genotype-Dependent Immunogenicity of Dengue Virus Type 2 Asian I and Asian/American Genotypes in Common Marmoset (Callithrix jacchus): Discrepancy in Neutralizing and Infection-Enhancing Antibody Levels between Genotypes"

_microorganisms, 2021, doi:10.3390/microorganisms9112196_

Round 1
Reviewer 1 Report
Azami et al. reported of differences in viral and immunological characteristics identified after marmoset infection with 2 different genotypes of DENV serotype 2. AI genotype induced longer viremia, but AA genotype resulted in higher virus titers in blood. Also, neutralizing antibodies induced by AA infection were detectable for a longer period of time. Interestingly, ADE of other serotypes was not observed for either animal group, while ADE of CM genotype was observed in AI infected animals. These results are very interesting and have important implications for what regards vaccine development. The study is well performed, methods are adequate and result interpretation is correct. My only concern is that I do not see a specific ethics committee approval. Was the approval of the Animal Study Review Board sufficient for this study?
Minor:
- Lines 31-2. Is this true? It doesn’t look like it from figure 5. From figure 5 it looks like only those animals inoculated with AI genotype showed ADE for CM genotype…
- Line 39. “Flaviviridae” needs to be italicized.
- Line 45. Homology is a binary concept (either something is or isn’t homologue) and degrees of homology do not exist. In these cases, the use of the word “identity” is preferable.
- Line 45. To which protein does this cut-off refer to?
- Line 47-8. Please specify the symptoms of these syndromes.
- Line 51. An infection can either be clinical or subclinical… what do you mean here?
- Lines 72-3. Viremia levels are not a characteristic of genotype-specific antibody levels. Please, rephrase the study objective.
- Lines 84-87. Please use uniform the virus names (e.g. DENV type 1 or DENV1).
- Lines 88-91. To which genotype do DENV2 DHF0663 and DENV2 TL-30 viruses belong? Also, there are 4 DENV-2 viruses listed here while only 3 are used in the study.
- Lines 99-101. Please add how inoculation was performed.
- Lines 110-115. Please add whether these kits are serotype specific or can detect antibodies against all serotypes.
- Line 138. Remove the comma before “resulting”.
- Line 142. Was this after the first or the second inoculation?
- Lines 193-196. I am wondering if you inverted the average detection period values for AI and AA since from the figure it looks to be the other way around. The same may be true for antigen detection duration. Please double check this paragraph.
Lines 209-10 and 319. I’d call them black and white bars rather than open and closed bars.
- Figure 2. Graph d looks different from the others as it misses the line on the top.
- Section 3.3. Which one was the Cosmopolitan genotype?
- Line 326. I don’t see the 51-fold increase in the figure. The mean seems below 32, while the highest value in the confidence range is almost 60…
- Line 369-71. Please check the grammar in this sentence.
- Line 378. The average value for the AA genotype is actually below the 4-fold increase.
- Lines 395-7. This sentence is redundant, and it can be removed.
- Lines 413-415. This sentence is not clear.
Reviewer 2 Report
The paper by Azami et al. is an interesting research article aiming at comparing the behaviour in terms of replication and antibodies
production of two different DENV2 genotypes in a monkey model.
About this, I have a major observation for the authors: why didn't they try to validate their results in a human model like human cell lines or sera from human infection cases?
I think that performing some viremia and antibody level assays in human patients, also archived sera, would be very much useful, if possible. Otherwise, the authors should make some considerations about
this point in the discussion/conclusions.
The introduction is well written though maybe a little too short. Some more information or reference could be provided about DENV pathogenicity
or about ADE, aimed to non-specialized readers. Also, if available, more updated epidemiological data would be welcome. And, maybe a table summarizing the classification
of DENV (at least upto the relevant genotypes).
The methods section is complexively good but some paragraph could be improved by adding some more information or verifying some unclear point (detailed in the 'minor points')
The results chapter has some weak points which are detailed as 'minor points', and in general should have some revision at least for the figures. The paragraph (3.4) about ADE,
which in my opionion is very important, is hard to follow, both for how it is written and for the figures, and should be substantially improved from both points to view.
Also, a better description of the related assay in the methods would be of much help. It would be much welcome if the authors could work on an improved version of this part of
the results.
The discussion-conlcusions are quite well written, but should highlight a little more that some of the results could be species-specific and that without further validation in humans it is hard
to give an interpretation to the obtained results.
Last note about the references: 10 articles out of 29 are self-citations: really an exagerated amount. The authors are kindly invited to reduce it to the strict necessary.
Some minor points:
- line 30: 'neutralizing antibody titer'
- line 52: ref.7 and 8 should be omitted (in that they simply requote ref.9), or moved elsewhere
- line 83: If I understood correctly, all viruses were isolated from the authors. In this case, please add information about the method of serotype-genotype classification
- line 102: the interval between inoculations is described as 4 months, but in many points of the paper there is a time schedule indicating that it was longer (more than 140 days, 4 months are 120-123 days)
- line 116: there are some details of this protocol about which I'm a little perplexed. For example, the use of plasma instead of serum, the very high infectious dose and
the use of 10% FBS to dilute the samples, and other things which are not sticky to the guidelines. Please verify.
- line 139: in this case also I have the similar doubts, being this assay strictly correlated with the previous one.
- line 156: please provide information about primers and reagents used for this assay.
- line 171: please write more correctly the way the viral titre is calculated
- line 191: it is not clear, nor explained in the methods, in which way has the duration been calculated. Please add details in one or both chapters.
By the way, fig.1 A shows a different trend, in that the AA group clearly looks to have more abundant antigen at day 10 pi.
- line 196: same as above:please explain how these time durations have been calculated. Furthermore, in fig.1b the AA group seems to have no viral production at day2,
which is quite not realistic. But, even in that case, it would be simply impossible to have a mean duration of 2 days......Please verify and correct.
- line 213: '....antibody levels'
- line 223: I think fig.2 would be more informative if unified in one (or two, divided by viral genotype) figures putting together the points relative to the three (or six) animals
- line 224: 'were still detected'
- line 241: the methods section makes no mention of the 'cosmopolitan' genotype. Please identify in par 2.1 which of the viruses is this one.
- line 244: fig.3b lacks the y axis. fig. 3a shows asterisks, but it i'm not sure it should. The higher antibody titre is in facts the non-specific one. I don't think in this case it should
be considered as a statistically significant result.
- line 288: the table descritpion should be below the table
- line 293: 'the virus titers.....' please rewrite in clearer way
- line 295: I think that fig 4 should be changed to a different kind of graphic; in this version it is difficult to follow, and not fully informative. It needs to include together all the samples for each virus
and use some different kind of graphic outfit to better describe the biological sense of the assay. Also, the descritpion should be unified and written more correctly (i.e., not 'fold infection-enhancement....', but something
more correct)
- line 390: this observation is not strongly supported by the results, as previously noted
Reviewer 3 Report
Azami et al. characterized in marmosets the viremia, neutralizing antibodies, and antibody-dependent enhancement induced by two DENV2 genotypes.
The following points may improve the manuscript:
- Figure 1a. I would suggest including in the figure that day 0-14 refers to second virus inoculation, as outlined in Figure 3.
Could you please provide how the ratio was calculated?
Did you apply a statistic test to compare the results from the two genotypes? Are those differences statistically significant?
I can speculate that the levels of NS1 in days 0-14 after the second inoculation were below the baseline because of the presence of antibodies specific to NS1; the authors may need to include an interpretation of that result.
Any particular reason not to have or test the infectious virus titer and level of virus RNA after the second inoculation?
- Figure 2. The resolution of the figure is poor. Also, consider coloring the graphs.
Again, I would recommend describing in the material and methods how the ratio was calculated and the reason to use ratio rather than original values.
The authors represented in the text the antibody results that are expected. Still, there is nothing about the differences or similitudes in the antibody responses induced by the AI or AA genotypes.
- Figure 3. What is the purpose of including the Cosmopolitan genotype in this figure? The authors may consider adding a sentence in the results section explaining the reason.
What represents each dot in the figure? Is it the result of each marmoset? Consider including the meaning in the legend.
The statistically significant differences seem to be between DENV2 AA compared to Cosmopolitan. However, the authors have to comment in the result section that none of the differences between the AI and AA were statistically significant.
If I am correct, each dot represents the result from each marmoset. In figure 3a, day 14 post-second infection, how did you compare one result from DENV2 cosmopolitan vs. DENV2 AA and how it was significant? I may be confused with this figure.
- Figure 4. The resolution of the figure is poor. It isn't easy to follow the results.
I do not see the fold infection-enhancement to the dengue virus type 4, and the authors need to include a sentence about that decision.
- Figure 5. I do not see the fold infection-enhancement to the dengue virus type 4.
Round 2
Reviewer 2 Report
The paper has been changed in some pars to meet good part of my observations. A few of my doubts have remained unsolved:
for example those regarding the neutralization assay protocol. The authors are invited to check for their protocol and provide a more inherent reply.
Another weak point was the exagerated amount of self citations, and this problem has been fixed only partially (there are now 8/31), I think the authors should go on to further refine their refeences selection.
Another point is about the antigen duration period: as I understand from your explanations, it is important to clarify also to the reader that the difference between genotypes is not in the duration of antigen production, but of antigen detectability. It is a critical difference when translating it into human diagnostics, so please change the text wherever necessary to highlight this point, also in the discussion
Reviewer 3 Report
Thank you to the authors for addressing the comments. In my opinion, the manuscript has been improved.
If possible, I would recommend the authors including a sentence about the limitations in the sample volume to test the DENV4 in figures 4 and 5.
